# Laboratory characterization of furan, 2(3H)-furanone, furaldehyde, 2,5-dimethylfuran, and maleic anhydride measured by PTR-ToF-MS

Wade Permar<sup>1</sup>, Mercedes Tucker<sup>1</sup>, and Lu Hu<sup>1</sup>

<sup>1</sup>Department of Chemistry and Biochemistry, University of Montana, Missoula, MT, USA Correspondence to: Wade Permar (Wade.Permar@mso.umt.edu)

Abstract. Furanoids are significant contributors to volatile organic compound hydroxyl radical reactivity in biomass burning emissions, yet their accurate measurement using proton-transfer-reaction time-of-flight mass spectrometry (PTR-ToF-MS) remains challenging due to potential interferences and measurement uncertainties. In this study, we conduct detailed laboratory characterizations of furan (C<sub>4</sub>H<sub>4</sub>O<sub>2</sub>, protonated m/z 69.033), 2(3H)-furanone (C<sub>4</sub>H<sub>4</sub>O<sub>2</sub>, m/z 85.028), 2-furaldehyde (C<sub>5</sub>H<sub>4</sub>O<sub>2</sub>, m/z 97.028), 2,5-dimethylfuran (C<sub>6</sub>H<sub>8</sub>O, m/z 97.065), and maleic anhydride (C<sub>4</sub>H<sub>2</sub>O<sub>3</sub>, m/z 99.008). Sensitivities for these compounds were found to have minimal dependence (less than 15 %) on both sample humidity and drift tube electric field strength (E/N). At high E/N (150 Td), fragmentation was observed for 2-furaldehyde (~8 %) at m/z 69.033, potentially interfering furan measurements, while hydrolysis products corresponding to m/z +18 ions were detected for 2(3H)-furanone, 2-furaldehyde, and maleic anhydride. The hydrolysis of maleic anhydride to maleic acid is most prominent, with the hydrate accounting for 7–31% of the parent ion signal across E/N conditions. Gas standard recertification confirmed the long-term stability of furanoids, and 21 other VOCs, in compressed gas mixtures, with changes in mixing ratios of less than 5 % over seven years, although PTR-ToF-MS instrument sensitivities decreased by ~30 % during this time, likely due to aging of the microchannel plate (MCP). While the stability of gas standards and the

minimal humidity and fragmentation effects support the accurate measurement of furanoids by PTR-ToF-MS, discrepancies with co-deployed gas chromatography-mass spectrometry highlight the need to further investigate potential isomeric and fragment interferences, particularly in aged BB smoke.

#### 1 Introduction

Furanoids, a class of volatile organic compounds (VOCs) consisting of a five-member ring with four carbons and one oxygen, are primarily emitted to the atmosphere from biomass burning (BB) (Akagi et al., 2011; Gkatzelis et al., 2024; Koss et al., 2018; Permar et al., 2021). Additional minor emission sources include volatile chemical products (McDonald et al., 2018), flavorings and food (Crews and Castle, 2007), vehicles (Wang et al., 2022), and biofuels (Román-Leshkov et al., 2007; Tuan Hoang and Viet Pham, 2021). Due to their high reactivity with OH and NO<sub>3</sub> radicals, furanoids can substantially

contribute to atmospheric oxidation capacity and influence ozone (O<sub>3</sub>) and secondary organic aerosol (SOA) formation 30 (Coggon et al., 2019; Joo et al., 2019; Romanias et al., 2024).

In wildfire smoke, furanoids constitute a significant OH sink, accounting for  $\sim$ 20 % of a smoke plume's initial OH reactivity and up to 10 % of near field O<sub>3</sub> production (Coggon et al., 2019; Gilman et al., 2015; Koss et al., 2018; Permar et al., 2023). Prominent contributors include C<sub>2</sub> substituted furans, 2-furaldehyde, furan, methyl furfurals, and furanone (Koss et al., 2018; Permar et al., 2023). Despite the atmospheric lifetime of many of these species being only a few hours, furanoids remain an important OH sink ( $\sim$ 10 % of plume OH reactivity) in smoke aged beyond three days due to the production of more oxygenated species such as 2-furoic acid/5-hydroxy-2-furfural (C<sub>5</sub>H<sub>4</sub>O<sub>3</sub>), succinic anhydride (C<sub>4</sub>H<sub>4</sub>O<sub>3</sub>), and maleic anhydride (C<sub>4</sub>H<sub>2</sub>O<sub>3</sub>)(Permar et al., 2021).

60

Furanoids are increasingly being used as BB tracers, particularly in mixed urban/smoke environments where acetonitrile can be less reliable (Bruns et al., 2017; Huangfu et al., 2021), Maleic anhydride is of particular interest due to its atmospheric lifetime of more than 5 days and it being formed primarily from furan oxidation (Coggon et al., 2019; Romanias et al., 2024). Shorter lived compounds such as furan and methyl furan are also useful indicators of smoke age and transport due to their lack of secondary production (Gkatzelis et al., 2024; Liang et al., 2022; O'Dell et al., 2020; Rickly et al., 2023; Robinson et al., 2021).

However, PTR-ToF-MS measurements of furanoids in fresh wildfire smoke have been observed to differ substantially from gas chromatography-based techniques, with PTR-ToF-MS reporting ~1.5–2 times more furan and up to 15 times more methyl furans than the gas chromatography-based Trace Organic Gas Analyzer (TOGA) (Gkatzelis et al., 2024; Permar et al., 2021). These discrepancies exist even after being corrected for known interferences based on laboratory characterization (Koss et al., 2018), suggesting that unresolved fragmentation or isomeric overlaps may still bias measurements.

While recent studies have expanded the list of detectable furanoids, sensitivities for most species are based on kinetically determined calibration factors with uncertainties up to 50 % (de Gouw and Warneke, 2007; Sekimoto et al., 2017). Furthermore, it is assumed that these species have little to no fragmentation in PTR-MS, which has not been rigorously tested for most furanoids. A recent review by Romanias et al. (2024) has extensively described the current state of the science for many atmospheric reactions of furanoids, but notes that PTR-ToF-MS "faces limitations in accurately quantifying furanoids, particularly in the presence of interfering compounds and under varying environmental conditions."

Resolving these challenges requires comprehensive laboratory investigations to elucidate the fundamental properties of furanoids and develop robust analytical methods for their detection and quantification."

Here, we present detailed laboratory characterizations of five atmospherically relevant furanoids: furan, 2(3H)-furanone, 2-furaldehyde, 2,5-dimethylfuran, and maleic anhydride. Our investigation encompasses the fragmentation, humidity dependence, and electric field-dependent sensitivities of these key VOCs in PTR-ToF-MS measurements. Additionally, we assess the long-term stability (7 years) of select furanoid gas standards. These results close important measurement gaps by showing that PTR-ToF-MS sensitives are largely unaffected by humidity or electric field conditions, and that gas standards can remain stable over multiple years. By ruling out these potential sources of bias, our findings indicate that remaining

discrepancies most likely stem from unresolved issues with speciation and interferences in complex airmass, highlighting the need for further identification work.

#### 65 2 Methods

## 2.1 Insturment description

Laboratory calibrations of furan ( $C_4H_4O$ , protonated m/z 69.033), 2(3H)-furanone ( $C_4H_4O_2$ , m/z 85.028), 2-furaldehyde ( $C_5H_4O_2$ , m/z 97.028), 2,5-dimethylfuran ( $C_6H_8O$ , m/z 97.065), and maleic anhydride ( $C_4H_2O_3$ , m/z 99.008) were carried out using a proton-transfer-reaction time-of-flight mass spectrometer previously described in detail by Permar et al. (2021).

Briefly, in PTR-ToF-MS, VOCs with proton affinities greater than water (691 kJ mol<sup>-1</sup>) are ionized via proton transfer reaction with H<sub>3</sub>O<sup>+</sup> reagent ions in the instrument's drift tube (R1). Most VOCs, including furanoids, have proton affinities greater than water, while the major constituents of air such as N<sub>2</sub>, O<sub>2</sub>, and CO<sub>2</sub> do not. Consequently, PTR-ToF-MS can measure hundreds to thousands of different VOCs in real time with no sample pretreatment at up to 10 Hz. The proton transfer reaction is also a soft ionization technique which typically results in little fragmentation of the analyte ions, allowing for them to be detected at their protonated mass (*m/z* +1). However, fragmentation can still occur, particularly with compounds that readily dehydrate or form stable intermediate cations (Coggon et al., 2024).

$$H_3O^+ + VOC \rightarrow VOC \cdot H^+ + H_2O (R1)$$

The efficiency of the proton transfer reaction is controlled by the drift tube energy, described as the E/N in Td. E/N is itself a product of drift tube temperature, pressure, and voltage, with higher E/N corresponding to increased collisional energy. Generally, higher E/N results in greater instrument sensitivity as more analyte is protonated and clustering with excess H<sub>2</sub>O is decreased (de Gouw and Warneke, 2007; Yuan et al., 2017). However, higher collisional energy can also cause more compounds to fragment, increasing the difficulty to identify and quantify the full mass spectrum. Optimal drift tube conditions then are a balance between getting sufficient sensitivities for analytes of interests while minimizing their fragmentation. As molecular structure, polarity, dipole, and functional groups all play a role in a compound's sensitivity and propensity to fragment, optimal E/N conditions can vary based on the analyte of interest.

In addition to sensitivity being dependent on instrument drift tube energy, the presence of excess water vapor in a sample can also impact the efficiency of the proton transfer reaction. Typically, the impact of humidity on instrument sensitivities is greatest for VOCs with proton affinities nearer to water, as the reverse reaction becomes more competitive (R1), leading to a decrease in measurement sensitivity. In PTR-MS, the ratio of water clusters to the primary ion ([H<sub>2</sub>O(H<sub>3</sub>O<sup>18</sup>)+]/[H<sub>3</sub>O<sup>18</sup>+] or m39/m21) is used as a proxy to estimate the humidity in the drift tube due to their linear relationship with relative humidity (de Gouw and Warneke, 2007; Stönner et al., 2016), and typically maintained at less than 5 %. Previous studies have shown the m39/m21 ratio can be used to derive humidity dependent sensitivities for VOCs with lower proton affinities (Baasandorj et al., 2015; Warneke et al., 2011a).

#### 2.2 Standard preparation and calibration system

105

95 Calibrations were conducted using a syringe pump with liquid standard prepared from furanoid standards diluted to 100–300 ppm in analytical grade ethyl acetate. Ethyl acetate was chosen for use as a solvent due to maleic anhydride and 2(3H)-furanone being reactive with water making it so neither water nor alcohols (which typically absorb water) could be used. Similarly, dimethylfuran is not miscible in water, while typical non-polar solvents such as cyclohexane and n-hexanes were found to have strong interference at m/z 69 due to either uncharacterized fragments and/or contamination in the solvent, even for commercially available analytical grade products.

Ethyl acetate has a proton affinity of 835.7 kJ mol<sup>-1</sup> and is therefore detectable by PTR-MS at *m/z* 89.060. When used as a solvent, ethyl acetate has high enough concentrations to saturate the instrument signal at *m/z* 89. The mass spectra of the pure ethyl acetate solvent also have notable peaks at *m/z* 43, 61, 79, and 107, likely due to fragmentation and contamination. Peaks from the <sup>13</sup>C and <sup>18</sup>O isotopes are also present. However, we found no overlap with the ethyl acetate solvent peaks and any of the furanoid compounds, nor any depletion of the primary ion. To account for shifts in the base line caused by the high ethyl acetate signal and presence of contaminate peaks, the instrument background was corrected across the range of solvent flows for each calibration. Additionally, raw counts are normalized to H<sub>3</sub>O<sup>+</sup> to account for variation in the instrument's primary ion signal and expressed here as normalized counts per second per ppb (NCPS ppb<sup>-1</sup>) (de Gouw and Warneke, 2007).

Standard additions were conducted using a custom-built syringe pump based calibration system (Fig. 1). The system works by injecting 50–150 nl min<sup>-1</sup> of liquid standard into 500 sccm of catalyst generated zero via a syringe pump (Harvard Apparatus PHD ULTRA) equipped with a 10 ul Hamilton gastight syringe. The standard is immediately evaporated into the zero air, and then further diluted and humidified in a heated chamber (100 °C) by mixing with 1000 sccm of zero air and 0–50 μl min<sup>-1</sup> of water, vaporized via a nebulizer. The PTR-ToF-MS inlet subsamples the humidified standard with excess flow going to the exhaust. Using this method, we can make sub-ppb standards, with most calibrations in this work carried out from 0.8–3 ppb.

Figure 1: Schematic of the syringe pump based calibration system used in this work.

The efficacy of the syringe pump system was validated by comparison to sensitivities determined from both compress gas standards and a commercially available Liquid Calibration Unit (LCU, Ionicon Analytik). Additionally, all syringe pump calibrations were carried out using acetonitrile as an internal standard at approximately the same concentrations as the furanoid of interest. For each furanoid calibration, the sensitivity for acetonitrile was also calculated and found to consistently be in good agreement with its sensitivity from compressed gas standard calibrations. Data when acetonitrile sensitivities varied by greater than 10 % were removed from further analysis, providing a per-sample check on the syringe pump's performance.

#### 2.3 Compressed gas standards

In addition to the calibrations done in this work using liquid standards, we also examined the stability of two seven-year-old compressed gas cylinders (Apel-Riemer Environmental, Inc.) that contain  $\sim$ 1 ppm each of furan, 2-furaldehyde, 2-methylfuran, and 5-methylfurfural, along with 21 other VOCs (Table A1). Both cylinders were originally prepared in 2017 and recertified by the manufacturer in 2024. Standards are gravimetrically prepared in cleaned, treated, high-pressure aluminum cylinders from traceable liquid standards. Accuracy is better than  $\pm$ 5 %, with GC-MS analysis confirming the accuracy of the gravimetric preparation and recertification concentrations. Cylinders are certified for 36 months, though many compounds are stable for several years.

#### 2.4. Fragmentation and E/N dependence

For each furanoid, the extent of fragmentation was first determined by identifying the major ion peaks at ~10 ppb of standard generated by the syringe system described above. Fragmentation was then probed at E/N of 110 Td and 150 Td, set by changing the drift tube voltage from 660 to 900 V while maintaining a constant temperature and pressure (60 °C, 3.0 mbar).

We limit our analysis to E/N 110–150 Td as we are no longer able to control the m39/m21 ratio within the desired 0–5 % outside this range. Instrument background was then determined using a constant injection of ethyl acetate followed by a constant injection of liquid standard. Humidity was controlled to maintain m39/m21 at 2.5–3.0 %. Fragments were then identified as masses with average normalized counts greater than three times the standard deviation of the background signal (3σ).

For all humidity dependent experiments, drift tube conditions were maintained at 810 V, 60 °C, and 3.00 mbar (E/N = 134 Td), which is the standard conditions used for our instrument during ambient sampling (Permar et al., 2021; Selimovic et al., 2022). Four-point calibrations were then conducted while adjusting the humidity of the carrier gas such that the m39/m21 ratio changed from 0.2 to 8 %, at ~0.5 % increments. Similarly, to determine how each species sensitivity depends on electric field strength, four-point calibrations were carried out at E/N 110–150 Td stepped at ~5 Td. This was done by

varying the electric field strength of the drift tube from 660–900 V while maintaining constant pressure and temperature. As the m39/m21 ratio is also dependent on the electric field strength, the humidity of the carrier gas was increased in proportion to the E/N to maintain m39/m21 at 2.5-3.0 % across all calibrations.

## 3 Results

#### 3.1 Fragmentation

Furan and dimethylfuran were both found to have minimal drift tube chemistry or fragmentation and are consequently detected at their parent ion masses across all E/N in this work (Table 1, Fig. 2). 2-furaldehyde, 2(3H)-furanone, and maleic anhydride were found to have notable fragmentation to ions other than the parent mass, especially at high E/N (Fig. 2). For 2-furaldehyde, at 110 Td only the parent ion is detected at *m/z* 97.028, while at 150 Td *m/z* 69.033 accounts for 17 % of the signal. This likely corresponds to a C<sub>4</sub>H<sub>4</sub>OH<sup>+</sup> ion forming from dehydration and subsequent fragmentation at the aldehyde group, similar to what has been observed in other long chain aldehydes (Coggon et al., 2024; Pagonis et al., 2019). This is of note as *m/z* 69.033 is commonly attributed to only furan (Koss et al., 2018; Pagonis et al., 2019; Yuan et al., 2017), meaning the fragmentation of 2-furaldehyde could bias PTR-ToF-MS furan measurements high in BB smoke where 2-furaldehyde and furan are emitted at similar rates (Koss et al., 2018; Permar et al., 2021). However, at our standard operating conditions (E/N 134, m39/m21 

Figure 2: Major ion masses detected when measuring furan, 2(3H)-furanone, 2-furaldehyde, 2,5-dimethylfuran, and maleic anhydride at E/N of 110 Td (blue bars) and 150 Td (orange bars). Each bar corresponds to measured ions with average normalized counts greater than 3σ the background signal.

Table 1. Summary of the humidity and electric field dependent sensitivity changes for the furanoids analyzed in this work, along with their major fragment ions as a proportion of their primary ion signal.

# Sensitivity changes as a product of:

| Species           | Formula                                      | Parent ion mass m/z | m39/m21<br>(0.2-8 %) | E/N (110-<br>150 Td) | Fragments<br>and other<br>products              | Fragment m/z | Percent of parent ion |
|-------------------|----------------------------------------------|---------------------|----------------------|----------------------|-------------------------------------------------|--------------|-----------------------|
| furan             | C <sub>4</sub> H <sub>4</sub> O              | 69.033              | -14 %                | -2 %                 | -                                               | -            | -                     |
| 2-(3H)furanone    | C <sub>4</sub> H <sub>4</sub> O <sub>2</sub> | 85.028              | -1 %                 | -7 %                 | $C_4H_6O_3H+$                                   | 103.039      | 0-7 %                 |
| 2 famildaharda    | C <sub>5</sub> H <sub>4</sub> O <sub>2</sub> | 97.028              | 12 %                 | -6 %                 | $C_4H_4OH^+$                                    | 69.033       | 0-17 %                |
| 2-furaldehyde     |                                              |                     |                      |                      | $C_5H_6O_3H^+$                                  | 115.039      | 0-5 %                 |
| 2,5-dimethylfuran | C <sub>6</sub> H <sub>8</sub> O              | 97.065              | 6 %                  | -7 %                 | -                                               | -            | -                     |
| maleic anhydride  | $C_4H_2O_3$                                  | 99.008              | -9 %                 | -10 %                | C <sub>4</sub> H <sub>4</sub> O <sub>4</sub> H+ | 117.018      | 7-31 %                |

# 3.2 Humidity and E/N dependent sensitivity

To investigate furanoid sensitivity dependence on humidity, calibrations were carried out with the m39/m21 humidity proxy varied from 0.2 % to 8 % by vaporizing water into the calibration gas stream (Section 2). Figure 3 shows that sensitivities for all five furanoid are minimally impacted by humidity within this range, with the sensitivity change generally within the calibration uncertainty of 10 %. 2(3H)-furanone sensitivity was the least impacted by humidity, changing by less than 1 %. Furan, 2-furaldehyde, 2,5-dimethylfuran, and maleic anhydride sensitivities change by 6–15 %, with furan and maleic anhydride sensitivities being negatively correlated with humidity while 2-furaldehyde and 2,5-dimethlyfuran are positively correlated.

Figure 3. PTR-MS instrument sensitivity (expressed as normalized counts per second per ppb) as a function of humidity in the drift tube (expressed as mass 39/mass 21 humidity proxy). When available, different calibration methods are colored corresponding to those done using compressed gas standard cylinders, Liquid Calibration Unit, or syringe pump.

Figure 4 shows how the sensitivities for the five furanoids are impacted by changing the instrument E/N from 110–150 Td at a narrow fixed m39/m21 range. Furan sensitivity is least dependent on E/N, decreasing by 2 % over the range investigated. This is consistent with it not having any major fragmentation (Fig. 2). Alternatively, 2(3H)-furanone, 2-furaldehyde, 2,5-dimethylfuran, and maleic anhydride sensitivities all decrease 5–10 % with higher E/N. For 2(3H)-furanone, 2-furaldehyde, and maleic anhydride this likely reflects the increased prevalence of the hydrated ion and fragmentation at higher E/N. Additionally, it is likely these furanoids are reactive with the H<sub>2</sub>O(H<sub>3</sub>O)<sup>+</sup> cluster (proton affinity 801 kJ mol<sup>-1</sup>, Yuan et al., 2016), which is more favorable at lower E/N conditions. Overall, the E/N has a minimal impact on the sensitivity of these five furanoids, with the variability again within the calibration uncertainty. However, it is notable that higher E/N may negatively affect the quantification of masses with interfering fragments from these species.

Figure 4: Instrument sensitivity for furan, 2(3H)-furanone, furaldehyde, 2,5-dimethylfuran, and maleic anhydride as a product of the drift tube electric field strength (E/N) from 110–150 Td.

#### 3.3 Stability of gas standard concentrations and their instrument sensitivities'

The recertification of two compressed gas cylinders (Apel-Riemer Environmental, Inc.) show that the mixing ratios in the tanks changed by less than 1 % in the seven years since the standards were made (Furanoids summarized in Table 2 with the remaining VOCs in Table A1). For all VOCs, the change is well within the uncertainty of the recertification analysis (±5 %) and demonstrates that these furanoids and other VOCs are remarkably stable in gas standards.

Table 2 also shows how the sensitivity of our PTR-ToF-MS has changed over those same seven years, with the sensitivities for the four furanoids in our gas standard decreasing from 32–60 %. This decrease in sensitivity is also present in the other species regularly calibrated using gas standards (Table A1). Given that mixing ratios for all species in the gas standards have changed less than 5 % over this period, the decreased instrument sensitivity likely reflects a wearing of the instrument; with sensitivity being especially high in the new factory-tuned instrument then leveling out as the electronics aged. Importantly, this seven-year period also corresponds with the lifetime of the multichannel plate (MCP) detector in this particular instrument, which was replaced shortly after the calibrations in this study. Although the exact mechanisms leading to a

systematic drift in instrument sensitivity are beyond the scope of this work, it reiterates the importance of performing calibrations during field deployments and revisiting them over an instrument's lifetime.

Table 2. Comparisons of long-term sensitivities and gas standard concentrations for the furanoids discussed in this work. Data from 2017 corresponds to when the PTR-ToF-MS and gas standard was new. Data from 2024 is after seven years of use and corresponds with the gas standard being recertified. Calculated sensitivities follow Sekimoto et al. (2017) using 2024 sensitivities, and are shown with their percent difference of 2024 calibrations factors. Note that empty values correspond to sensitivities for species that are not present in the gas standards.

|                   | Gas standard concentration [ppb] |      |             | Sensitivity [NCPS/ppb] |      |       | Sensitivity calculated per Sekimoto et al. |  |
|-------------------|----------------------------------|------|-------------|------------------------|------|-------|--------------------------------------------|--|
| Species           | 2017                             | 2024 | 024 diff. 2 |                        | 2024 | diff. | (2017) ( % diff.)                          |  |
| furan             | 1022                             | 1029 | 1 %         | 10.9                   | 6.7  | -39 % | 5.8 (-13 %)                                |  |
| 2-methylfuran     | 978                              | 1009 | 3 %         | 10.9                   | 7.4  | -32 % | 6.3 (-15 %)                                |  |
| 2-(3H)furanone    | -                                | -    | -           | -                      | 13.1 | -     | 7.5 (-42 %)                                |  |
| 2-furaldehyde     | 1000                             | 1003 | 0 %         | 16.2                   | 9.5  | -41 % | 12.3 (30 %)                                |  |
| 2,5-dimethylfuran | -                                | -    | -           | -                      | 5.9  | -     | 6.8 (16 %)                                 |  |
| maleic anhydride  | -                                | -    | -           | -                      | 8.7  | -     | 9.8 (13 %)                                 |  |
| 5-methylfurfural  | 977                              | 999  | 2 %         | 17.2                   | 6.9  | -60 % | 11.1 (61 %)                                |  |

Detailed laboratory calibrations are often time-consuming and impractical to obtain for the hundreds of different VOCs measured by PTR-ToF-MS. However, one major benefit of PTR-MS is that sensitivities can be readily calculated using known drift tube parameters (i.e., pressure, temperature, voltage, and length) and each VOC's proton transfer reaction rate with H<sub>3</sub>O<sup>+</sup> (k<sub>PTR</sub>) (de Gouw and Warneke, 2007). A notable limitation of this approach is that well defined k<sub>PTR</sub> values are needed for each VOC of interest, which is often not the case for newly identified species, including all furnaoids in this work except furan and 2-furaldehyde.

In the absence of known k<sub>PTR</sub> values, Sekimoto et al. (2017) showed that proton transfer reaction rates could be estimated using molecular properties (k<sub>cap</sub>). As k<sub>PTR</sub> has a linear relationship with instrument sensitivity (Cappellin et al., 2012; Sekimoto et al., 2017), the slope of the regression between k<sub>PTR</sub>, and by extension k<sub>cap</sub>, versus sensitivities measured from direct calibration can then be used to calculate sensitivities for unknown VOCs. For the instrument used in this work, the slope of k<sub>cap</sub> vs. sensitivity is 3.3x10<sup>9</sup> with an r<sup>2</sup> of 0.93. The accuracy of this approach is estimated to be less than 50 %, determined by the difference between the estimated sensitivities and the calibrated ones for selected VOCs (Permar et al., 2021; Sekimoto et al., 2017; Selimovic et al., 2022).

Table 2 shows how the furanoid sensitivities from calibrations in this work compare to those calculated based on the method described by Sekimoto et al. (2017). Table A1 shows the same for the other 21 VOCs in our gas standards. For the furanoids reported here, we find that direct calibrations for all furanoids except 5-methylfurfural agree within less than 42 % of those

calculated from their molecular properties. The reason for 5-methylfurfural's calculated sensitivity being 60% lower than its measured is unknown, though it could in part be due to reaction with the water dimer which is not accounted for in these calculations. This indicates that in the absence of direct furanoid calibrations, their kinetically calculated sensitivities provide a good estimate of the instrument response. We also find that the slope of the correlation between k<sub>cap</sub> and the measured sensitivities for 25 directly calibrated VOCs decreased by 33 % in the seven years from 2017 to 2024, representative of the overall decrease in instrument sensitivity described above. Consequently, calibration factors calculated based on the correlation of k<sub>PTR</sub> vs. sensitivity of directly calibrated species should also be revisited regularly to account for changes in instrument response.

#### 3.4 Furanoid speciation and potential fragment interference in PTR-ToF-MS

The lack of major fragmentation and humidity dependence as described above demonstrates that PTR-ToF-MS furanoid measurements are robust under standard sampling conditions. Additionally, the long-term stability of key furanoids in calibration standards means that instrument sensitivities should be well quantified with regular calibrations. Consequently, the discrepancy between PTR-ToF-MS and GC-MS furanoid measurements in BB smoke cannot be explained by calibration error alone and is instead likely due to the presence of unidentified interfering fragments and/or isomers. Extensive work has been done to identify PTR-ToF-MS measured masses in BB smoke, relying on both direct speciated measurements and association with compounds known to be in smoke from literature review.

Table 3 shows potential interfering isomers in BB smoke for the furanoid masses calibrated here, as well as three others we regularly calibrate via their gas standards. Importantly, although each of these isomers have been reported to be present in BB smoke, little work has been done to determine their fractional contributions. Additionally, the extent of interference from fragments of high order masses is largely unknown. This is due to the difficulty in making such speciated measurements in complex airmasses, where PTR-ToF-MS measurements would need to be done in-line with a separation technique such as gas chromatography.

For fresh BB emissions, one of the most complete studies reporting speciated furanoid measurements from PTR-ToF-MS was done by Koss et al. (2018) using laboratory burning experiments and a combination of online GC-MS measurements, literature review, and correlation analysis. In Table 3 we report the fractional distribution for each of the masses calibrated in this work as reported by Koss et al., (2018). Additional fractional speciation is reported from Permar et al., (2021) and Gkatzelis et al., (2024) where available. We note that the fractional contributions from Koss et al., (2018) are based on very fresh smoke measured directly above a laboratory burn, while those from Permar et al., (2021) and Gkatzelis et al., (2024) represent wildfire emissions measured by aircraft 10-120 minutes downwind from the fire and calculated from co-deployed GC-MS measurements made by TOGA (Apel et al., 2003, 2010, 2015; Hornbrook et al., 2011).

As demonstrated in Table 3 and by measurement comparisons between PTR-ToF-MS and GC-MS, large discrepancies exist in the speciation of masses attributed to furan, methyl furans, and furaldehydes in BB smoke. For example, *m/z* 69.033 is attributed to just furan in very fresh laboratory BB emissions, while only 46-66 % is attributed as furan in wildfire plumes

sampled 20-120 minutes down wind of their source. Similarly, methyl furans at m/z 83.049 and furaldehydes at m/z 97.065 also have a higher unknown fraction in the field measurements. This may be due to the rapid change in smoke composition as it ages post emission, with unidentified isomers or fragments being formed.

This variability highlights that as BB plumes age, isomers and fragments can significantly contribute to the signals of some furanoids. Such processes complicate PTR-ToF-MS interpretation of these masses in aged smoke, urban environments, and mixed smoke/urban air masses where little to no detailed speciation has been done. While laboratory and field studies suggest that furanoids are reliable BB tracers, in the absence of complimentary measurements, their accuracy in aged smoke may be less robust. Consequently, more detailed studies utilizing co-deployed GC and PTR-MS are needed to better measure furanoids in varying real-world conditions.

Table 3. Potential interfering isomers in fresh BB smoke for furanoid masses discussed in this work. Known isomers that have been reported in BB smoke are listed, though most are from studies without co-deployed PTR-MS. Fractional contributions are for each isomer to the total PTR-ToF-MS signal at that mass where available. Values from Koss et al. (2018) are from laboratory burning experiments and a combination of online GC-MS measurements, literature review, and correlation analysis. Fractions for Permar et al., (2021) and Gkatzelis et al., (2024) represent wildfire emissions measured by aircraft 10-120 minutes downwind from the fire and are calculated from co-deployed GC-MS measurements made by TOGA. Unidentified fractional contributions represent the remaining portion of the instrument signal not attributed to the corresponding known isomers at that mass.

#### Fractional contribution in smoke

| Ion mass m/z | Formula                                      | Isomers known to be in BB smoke from largest to smallest contribution | Koss et al., (2017) | Gkatzelis et al.,<br>(2024) | Permar et al., (2021) |  |
|--------------|----------------------------------------------|-----------------------------------------------------------------------|---------------------|-----------------------------|-----------------------|--|
| 69.033       | C <sub>4</sub> H <sub>4</sub> O              | furan <sup>2–4,6,11,12,19,21</sup>                                    | 100 %*              | 46 %                        | 66 %                  |  |
|              |                                              | 2-methylfuran <sup>2-4,7,9,11,12,18,19,21,22</sup>                    | 51 %                | 15 %                        | 16 %                  |  |
|              |                                              | 3-Methylfuran <sup>4,7,11,12</sup>                                    | 10 %                | 3 %                         | 3 %                   |  |
| 83.049       | C <sub>5</sub> H <sub>6</sub> O              | cyclopentenones <sup>4,7</sup>                                        |                     |                             |                       |  |
|              | C)1100                                       | Likely minor:                                                         | 37 % unidentified   | 82 % unidentified           | 81 % unidentified     |  |
|              |                                              | Pent-2-ynal <sup>7</sup>                                              | 37 70 diffacilitied |                             |                       |  |
|              |                                              | 1,4-Pentadien-3-one <sup>7</sup>                                      |                     |                             |                       |  |
| 85.028       | $C_4H_4O_2$                                  | 2-(3H)furanone <sup>1,2,8,12,14,17,19</sup>                           |                     |                             |                       |  |
|              |                                              | 2-furaldehyde <sup>1,4,7–9,12–14,18,19</sup>                          | 84 %                | 22 %                        |                       |  |
| 97.028       | $C_5H_4O_2$                                  | 3-Furaldehyde <sup>2,4,7,11,12,19,21</sup>                            | 4 %                 | 2 %                         |                       |  |
|              |                                              | cyclopentenediones <sup>4,7,22</sup>                                  | 11 % unidentified   | 76 % unidentified           |                       |  |
|              |                                              | 2,5-dimethylfuran <sup>2–4,7,11,12,18,19,22</sup>                     | 44 %                |                             | ·                     |  |
|              |                                              | 2-Ethylfuran <sup>2,4,7,11,19,22</sup>                                | 10 %                |                             |                       |  |
| 97.065       | $C_6H_8O$                                    | 2-Methylcyclopentenone <sup>6,13,14,17</sup>                          |                     |                             |                       |  |
|              |                                              | 2,4-dimethylfuran <sup>7</sup>                                        | 46 % unidentified   |                             |                       |  |
|              |                                              | 2,3-dimethylfuran <sup>7</sup>                                        |                     |                             |                       |  |
| 99.008       | C <sub>4</sub> H <sub>2</sub> O <sub>3</sub> | maleic anhydride <sup>3,15</sup>                                      |                     |                             |                       |  |
|              |                                              | 5-methylfurfural <sup>3,7,10,12,14,17–19</sup>                        |                     |                             |                       |  |
| 111.044      | $C_6H_6O_2$                                  | Catechol (Benzenediols) 3,4,8,12,13,17–20                             |                     |                             |                       |  |
|              |                                              | 2-acetylfuran <sup>7</sup>                                            |                     |                             |                       |  |

References: <sup>1</sup>Azeez et al., 2011; <sup>2</sup>Brilli et al., 2014; <sup>3</sup>Bruns et al., 2017; <sup>4</sup>Gilman et al., 2015; <sup>5</sup>Gkatzelis et al., 2024; <sup>6</sup>Hatch et al., 2015; <sup>6</sup>Hatch et al., 2019; <sup>7</sup>Heigenmoser et al., 2013; <sup>8</sup>Ingemarsson et al., 1998; <sup>9</sup>Jordan and Seen, 2005; <sup>10</sup>Karl et al., 2007; <sup>11</sup>Koss et al., 2017; <sup>12</sup>Li et al., 2013; <sup>13</sup>Liu et al., 2012; <sup>14</sup>Müller et al., 2016; <sup>15</sup>Permar et al., 2021; <sup>16</sup>Pittman et al., 2012; <sup>17</sup>Simmleit and Schulten, 1989; <sup>18</sup>Stockwell et al., 2015; <sup>19</sup>Veres et al., 2010; <sup>20</sup>Warneke et al., 2011b; <sup>21</sup>Yokelson et al., 2013.

\*From eight laboratory burns, four of which the GC shows many small peaks with furan being less than 3 % of the signal, while the other four attribute the full signal to furan.

#### 315 4 Conclusion

320

325

Detailed laboratory characterization of furan ( $C_4H_4O$ , m/z 69.033), 2(3H)-furanone ( $C_4H_4O_2$ , m/z 85.028), 2-furaldehyde ( $C_5H_4O2$ , m/z 97.028), 2,5-dimethylfuran ( $C_6H_8O$ , m/z 97.065), and maleic anhydride ( $C_4H_2O_3$ , m/z 99.008) demonstrates that their sensitivities in PTR-ToF-MS have minimal humidity and drift tube electric field strength (E/N) dependence (<15 %). Furan and 2,5-dimethylfuran were found to not fragment upon proton transfer reaction. However, 2-furaldehyde was found to fragment at ~8 %, under typical operation conditions (E/N 130 Td), to m/z 69.033, interfering with the measurement of furan. A hydrolysis product was also observed for 2(3H)-furanone, 2-furaldehyde, and maleic anhydride, corresponding to their m/z+18 ion mass. For 2(3H)-furanone, 2-furaldehyde this product is minor, accounting for < 7 % of their parent mass at E/N 150 Td. The hydrolysis of maleic anhydride to maleic acid is more favorable, ranging from 7–31 % of the parent ion signal over the E/N range of 110-150 Td. Although we do not directly calibrate succinic anhydride in this work, we hypothesize that it will also undergo a similar hydration reaction as maleic anhydride in the PTR-MS.

Recertification of seven-year-old gas standard mixtures also showed that furanoids and other VOCs are very stable in compressed gas cylinders, with most mixing ratios changing by less than 1 % over the period. Over the same period average instrument sensitivities for the PTR-ToF-MS used in this work were found to decrease by ~33 %. This likely represents a settling in of the instrument electronics from its factory new condition, along with the aging of the MCP.

The stability of the furanoid gas standards, coupled with the minimal fragmentation and humidity dependence indicates that with regular calibrations PTR-ToF-MS measures furanoids accurately. Despite this, mixing ratios reported by PTR-ToF-MS measurements have been found to be significantly higher than those made by co-deployed GC-MS, indicating the presence of interfering isomers and/or fragments in PTR-ToF-MS. The full extent of these interferences is currently difficult to quantify because their contributions cannot be fully resolved without chromatographic separation, which decreases temporal resolution while adding instrument and analysis complexity. Consequently, improving the accuracy of furanoid measurements made by PTR-MS will require co-deployed GC-MS and/or GC-PTR-ToF-MS configurations to identify fragment and isomer interferences in complex air mixtures, including aged BB smoke and urban environments.

# Appendix A

Table A1. Comparisons of long-term sensitivities and gas standard concentrations for 21 additional VOCs discussed in this work.

Data from 2017 corresponds to when the PTR-ToF-MS and gas standards were new. Data from 2024 is after seven years of use

and corresponds with the gas standards being recertified. Calculated sensitivities follow Sekimoto et al. (2017) using 2024 sensitivities and are shown with their percent difference of 2024 calibrations factors.

|                                |      | as standaro<br>ntration [p |       | Sensitivity [NCPS/ppb] |      |       | Sensitivity calculated per Sekimoto et al. |  |
|--------------------------------|------|----------------------------|-------|------------------------|------|-------|--------------------------------------------|--|
| Species                        | 2017 | 2024                       | diff. | 2017                   | 2024 | diff. | (2017) (% diff.)                           |  |
| Methanol                       | 1022 | 1010                       | -1 %  | 16.3                   | 10.6 | -35 % | 7.4 (-30 %)                                |  |
| Propyne                        | 1007 | 993                        | -1 %  | 9.4                    | 7.1  | -24 % | 5.4 (-25 %)                                |  |
| Acetonitrile                   | 1016 | 1024                       | 1 %   | 21.6                   | 15.0 | -30 % | 12.8 (-15 %)                               |  |
| Acetaldehyde                   | 1022 | 995                        | -3 %  | 18.9                   | 12.5 | -34 % | 10.0 (-20 %)                               |  |
| Ethanol                        | 1019 | 1032                       | 1 %   | 0.9                    | 0.7  | -26 % | 7.4 (999 %)                                |  |
| 1-butene                       | 1003 | 999                        | 0 %   | 6.2                    | 4.3  | -30 % | 5.8 (34 %)                                 |  |
| Acetone                        | 972  | 977                        | 1 %   | 20.1                   | 14.9 | -26 % | 10.4 (-30 %)                               |  |
| Dimethyl sulfide               | 1000 | 992                        | -1 %  | 12                     | 9.2  | -24 % | 6.9 (-24 %)                                |  |
| Isoprene                       | 989  | 982                        | -1 %  | 7.3                    | 4.8  | -34 % | 6.3 (32 %)                                 |  |
| Methyl vinyl ketone            | 971  | 993                        | 2 %   | 15.9                   | 10.1 | -36 % | 10.6 (5 %)                                 |  |
| Methacrolein                   | 1007 | 996                        | -1 %  | 11.1                   | 7.3  | -34 % | 10.6 (45 %)                                |  |
| Methyl ethyl ketone            | 1011 | 1049                       | 4 %   | 18.1                   | 12.1 | -33 % | 10.3 (-15 %)                               |  |
| Benzene                        | 999  | 1006                       | 1 %   | 9.4                    | 6.9  | -26 % | 6.4 (-8 %)                                 |  |
| Toluene                        | 994  | 1002                       | 1 %   | 10.6                   | 7.3  | -31 % | 6.8 (-7 %)                                 |  |
| 3-hexanone                     | 943  | 941                        | 0 %   | 13.5                   | 7.6  | -44 % | 10.4 (36 %)                                |  |
| Ethylbenzene                   | 1003 | 995                        | -1 %  | 5.3                    | 3.4  | -36 % | 7.3 (115 %)                                |  |
| <i>m</i> -xylene               | 990  | 975                        | -2 %  | 9.5                    | 6.7  | -30 % | 7.3 (10 %)                                 |  |
| 1,2,4-trimethylbenzene         | 999  | 988                        | -1 %  | 11.3                   | 6.4  | -44 % | 7.6 (19 %)                                 |  |
| 1,3,5-trimethylbenzene         | 989  | 988                        | 0 %   | 11.3                   | 7.1  | -37 % | 7.6 (7 %)                                  |  |
| 1,2,3,5-<br>tetramethylbenzene | 997  | 997                        | 0 %   | 11.5                   | 6.9  | -40 % | 8.0 (16 %)                                 |  |
| α-pinene                       | 967  | 954                        | -1 %  | 5.1                    | 2.8  | -45 % | 8.1 (186 %)                                |  |

# 345 Code availability

The code related to this article is available upon request to the corresponding author.

# Data availability

The data related to this article is available upon request to the corresponding author.

#### **Author contribution**

WP: conceptualization, methodology, investigation, data curation, formal analysis, and original draft preparation. MT: investigation, methodology, review and editing. LH: conceptualization, funding acquisition, supervision, review and editing.

## **Competing interests**

The authors declare that they have no conflict of interest.

# Acknowledgments

This study was supported by the US National Science Foundation (AGS-2144896 and EPSCoR-2242802). The authors acknowledge the use of OpenAI's ChatGPT for assistance with light editing, including writing clarity, grammar, and summarization during the preparation of this manuscript. All results, analysis, and interpretations are solely those of the authors.

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
