# Peer review of "Laboratory characterization of furan, 2(3H)-furanone, furaldehyde, 2,5-dimethylfuran, and maleic anhydride measured by PTR-ToF-MS"

_EGUsphere, 2025_

## Author Response (AR1)

**Reviewer 1**

**Comments:**

This manuscript systematically validates the capabilities and limitations of PTR-ToF-MS in measuring furanoids, highlighting the need to account for fragmentation and isomeric interferences in biomass burning (BB) research. The findings provide essential data and guidance for improving the accuracy of VOC measurements. The results presented are insightful and are expected to be useful to the PTR-MS and BB communities. While the study presents a novel and valuable research angle, the manuscript's structure (e.g., Methods) requires significant refinement. I recommend its publication to *AMT*, after the authors have addressed the following comments.

We thank the reviewers for their thoughtful and constructive feedback. We have carefully considered each of the referee's comments and have revised the manuscript accordingly. Our detailed responses to each point are provided in bold type below each comment.

1. For introduction, several issues in structure, clarity, and conciseness need to be addressed to improve readability and scientific communication. My detailed comments are as follows:

**1.1 Organization and Logical Flow**

The Introduction attempts to cover a broad range of information, but its current structure is overly dense and lacks a clear logical progression. Key ideas are often repeated or presented out of sequence.

**1.2 Study Objective and Scope**

The study objectives are mentioned twice, in two separate places, with some redundancy. It would be more effective to consolidate this into a single clear statement toward the end of the Introduction. Additionally, the last paragraph should more directly explain how the work addresses the identified gaps.

**1.3 Redundancy and Repetition**

Several concepts are repeated unnecessarily. For example, the dominant role of biomass burning as a source of furanoids is stated in multiple places with similar wording. Similarly, the discussion of specific furanoids and their atmospheric roles appears both before and after the study objective is introduced. These points should be consolidated for conciseness.

We thank the reviewer for these detailed suggestions. We have rewritten the introduction to improve logical flow, reduce redundancy, and consolidate the objectives into a single, clear section near the end. The revised version also more directly explains how our study

addresses key measurement gaps. We believe these changes significantly improve readability and scientific communication of the manuscript.

2. For the Methods section, It is recommended to divide the Methods section into several subsections for better clarity and readability. The details of standard gas should be shown here but not in results.

This is a good point, and we have revised the methods section into subsections to improve clarity and readability. In addition, details of the compressed gas standards have been moved from the results to the methods section as recommended.

3. The results shown in Figure 2 demonstrate that the fragmentation of certain furanoid species can vary dynamically with changes in E/N. Therefore, readers may also expect to see a plot of fragmentation ratios as a function of E/N.

We thank the reviewer for this suggestion. While we did consider including a plot of fragmentation ratios as a function of E/N, we ultimately decided not to because the trends are not particularly informative for most species. Maleic anhydride shows the most notable fragmentation dependence on E/N, and this is already described and quantified in the text. For the other species, the fragmentation ratios remain essentially constant, making such a figure redundant and was therefore excluded to limit duplicating results.

4. The y-axis in Figures 3 and 4 is labeled in Ncps, which may make it difficult for readers to interpret the relative changes in signal intensity. Using relative signal values—for example, normalizing the maximum to 1—would improve clarity.

We have clarified in the methods section that NCPS (normalized counts per second) refers to the PTR-MS counts normalized to the primary ion signal, H3O+. This is standard practice in analyzing PTR-MS data to account for variation in the primary ion. Although we agree that normalizing to the maximum sensitivity could show the variation relative to each species more clearly, it would in turn make comparisons of sensitivities across instruments impractical.

5. Can you also provide the relationship of the Reagent ion signals and m39/m21 as a function of water vapor mixing ratios in the instrument? This can help clarify for readers the range of water vapor concentrations corresponding to your humidity experiments.

We appreciate this suggestion. However, we did not directly measure water vapor mixing ratios during these experiments for two reasons. First, the m39 (H3O18(H2O)+) signal is not solely determined by sample humidity, but is also strongly affected by instrument settings such as the efficiency of water vapor removal between the ion source and drift tube, as well as E/N. For example, the m39 signal can be changed by adjusting the instruments source valve, which is regularly done to optimize the m39/21 ratio for different sampling conditions. Consequently, the most accurate estimate of water vapor mixing ratios would require direct measurement in the drift tube, which is not practical. Second, prior work has shown that, under fixed instrument settings, the m39 and m21 scale linearly with relative humidity and is widely accepted as a proxy for quantifying humidity dependence (de Gouw and Warneke, 2007; Stönner et al., 2016).

6. Lines 228-230: "For most species, the mixing ratios in the tank changed by less than 1 % in the seven years since the standard was made (Furanoids summarized in Table 2 with the remaining VOCs in Table A1)." Please clarify the details of how the absolute concentration in the tank was measured for 2017 and 2024, including the measurement methods used.

Thank you for pointing out that this information was missing from the original manuscript. The initial tank concentrations and recertification was done by the tank manufacturer Apel-Riemer Environmental, Inc. Per the tank certification documents, concentrations are determined using GC-MS; "We prepare each cylinder individually. Accuracy is better than +/- 5%. Analysis confirms the accuracy of the gravimetric preparation. We use a series of NIST, NIST-traceable, NPL, and in-house gravimetric standards to perform the instrument calibrations."

We have added this information to the methods section.

7. Lines 228-230: "Table A1 shows the same for the other 21 VOCs in our gas standards. For the furanoids reported here, we find that direct calibrations for all furanoids except 5-methylfurfural agree within less than 42 % of those calculated from their molecular properties." Please provide the kPTR values used in your sensitivity estimations. It is also recommended to discuss the possible factors that may have contributed to the large discrepancy between the estimated and measured values for 5-methylfurfural.

We have added the slope of  $k_{PTR}$  vs known instrument sensitivities used to estimate calibration factors. We have also clarified that in the absence of a known  $k_{PTR}$  value, as is the case with most of the compounds discussed in this work,  $k_{PTR}$  is estimated using molecular properties, and more accurately referred to as  $k_{cap}$ . Specifically, "For the instrument used in this work, the slope of  $k_{cap}$  vs. sensitivity is  $3.3x10^9$  with an  $r^2$  of 0.93".

As per the 60% difference in the calculated vs measured 5-methylfurfural, we do not know the exact reason for this discrepancy but as pointed out by the other referee, one possible explanation is that the furanoids are likely also reactive with the water cluster (H2O)H3O+. As the kcap relationship assumes reaction only with H3O+, the presence of significant water cluster reactions could in part explain these discrepancies. We have added this to our discussion in the manuscript.

8. Lines 264-266: "We also find that the correlation between kptr and the measured sensitivities for 25 directly calibrated VOCs decreased by 33 % in the seven years from 2017 to 2024, representative of the overall decrease in instrument sensitivity described above." The correlation (R) between kptr and sensitivity should not change simply because the overall sensitivity decreases. Does 'correlation' in this context refer to the slope between the two variables or the correlation coefficient (R)? If the correlation (R) indeed decreases with a reduction in general sensitivity, please provide further explanation.

Thank you for pointing out that this statement is unclear. The original text was meant to communicate that the slope between the kptr and sensitivities decreased by 33%, not the correlation coefficient, which remains unchanged. We have clarified the manuscript.

9. Lines 295-297: "Similarly, methyl furans at m/z 83.049 and furaldehydes at m/z 97.065 also have a higher unknown fraction in the field measurements. This may be due to the rapid change in smoke composition as it ages post emission, with unidentified isomers or fragments being formed." Considering that the variability in fragmentation fractions may result from rapid changes in smoke composition, could the authors elaborate on how this might impact measurements in ambient air? Are there any ambient observations (e.g., urban environment but not wildfire) data available to support this? In light of the potential interferences, is furan—or other furanoid species—still a reliable tracer for biomass burning?

This is an important point, and we have expanded the discussion in the results section to better address how changes in smoke composition could influence PTR-ToF-MS measurements of furanoids. Specifically, we note that fragments and isomers may decrease the accuracy of PTR furanoid measurements in aged smoke.

**Other comments:**

1. The reference lists in several sentences are extensive and somewhat overwhelming. In some cases, citations could be streamlined or selected more selectively. Additionally, references should be consistently ordered chronologically, e.g., (Akagi et al., 2011; Stockwell et al., 2015; Koss et al., 2018; Andreae, 2019; etc.)

We acknowledge the balance between readability and thorough citation of relevant work. We have reviewed the longer citation lists and streamlined them where appropriate. Regarding the order of in-text citations, formatting was handled through AMT's Zotero template, which defaults to alphabetical order. AMT guidelines state that in-text citations may be ordered by relevance, chronology, or alphabetically, depending on the author's preference. As such, and to maintain consistency with the formatting template, we have elected to retain the alphabetical ordering.

2. Please unify 2,5-dimethylfuran or 2,5-dimethyl furan.

All instances have been changed to 2,5-dimethylfuran.

3. Line 41 Typos for 2-furalehdye

Fixed.

4. Lines 52 "photochemical chemical aging" should be photochemical aging.

Fixed.

5. **Table 3** What does 'unknown' refer to in your table? Please clarify its meaning in the text.

We have changed unknown to unidentified for better clarity. Also, we have added the following to the table caption: "Unidentified fractional contributions represent the remaining portion of the instrument signal not attributed to the corresponding known isomers at that mass."

6. Line 172 Typos for malic anhydride

**Fixed.**

7. **Lines 324-326** It is recommended to specify whether "higher" refers to concentration, sensitivity, or signal intensity to avoid ambiguity.

We have clarified that this statement is about mixing ratios.

8. Line 326 "The full extent of these interferences is currently difficult to quantify" It can be helpful to state the specific reasons for this difficulty.

We have revised this sentence to better clarify these difficulties. Specifically, isomer/fragment "contributions cannot be fully resolved without chromatographic separation, which decreases temporal resolution while adding instrument and analysis complexity'

**Reviewer 2**

Permar et al. present an analysis of PTR-ToF-MS responses to furanoid compounds to address reported uncertainties associated with quantifying furanoids in ambient biomass burning smoke. Furanoids are important VOCs emitted in wildfire smoke that have been well-quantified by PTR-ToF-MS in laboratory settings (e.g., Koss et al. 2018), yet work by Permar et al. 2021 and Gkatzelis et al. 2024 show that field observations of furanoids have large discrepancies relative to GC-MS techniques. This difference could be associated with challenges in PTR-ToF-MS detections of furanoids, detection of previously understudied isomers, or instrument interferences associated with fragmentation. This paper evaluates the PTR-ToF-MS response to furanoids to determine detection robustness.

Overall, the authors show that furanoids are well quantified by PTR-ToF-MS and that detection is not strongly impacted by instrument operating conditions, including relative humidity and E/N. Furthermore, furanoids exhibit remarkable stability in certified calibration standards, implying that long-term monitoring can be achieved by regular instrument calibration. These results demonstrate that field discrepancies are not likely due to instrument detection issues, but rather to fragmentation interferences or detection of unknown isomers. More work is needed to quantify product ions in ambient smoke using GC-preseparation, or other separation techniques.

The manuscript is well-written and an important contribution towards quantifying a key VOC class emitted from wildfire smoke. I recommend publication and only have minor comments that the authors might consider.

We thank the reviewer for their thoughtful and positive evaluation of our work. Our responses to individual comments can be found below in bold type.

Line 15: This statement might be confused with the stated impact of humidity dependencies at line 12. Perhaps reiterate that this variability in hydrolysis is associated with changes in E/N rather than humidity.

We have edited this statement to make it clearer that we are discussing changes in E/N.

Line 86 - 87, 100-103: This section is a very nice description of PTR. It would be helpful to include some equations to inform readers who may not be familiar with hydration or C-C fragmentation in PTR-MS.

Thank you for the suggestion. We have chosen not to add additional equations in this section, as our focus is on describing general instrument behavior rather than detailed reaction mechanisms. We feel that including and explaining these equations in a robust way would distract from our main point. However, we have added a citation to a recent paper (Coggon et al., 2024) with an excellent discussion of these fragmentation mechanisms.

Line 113-117: This is great information. Is ethyl acetate non-fragmenting? Likewise, do any of the isotopes pose any interferences of concern? I think this is useful to note for others who may consider ethyl acetate as a solvent.

This is an excellent question, and we have updated the manuscript to include the most prominent additional peaks seen when measuring the 'pure' ethyl acetate. Notable, we see enhancements at m/z 43, 61, 79, and 107, likely due to fragmentation and contamination. Peaks from the  $^{13}$ C and  $^{18}$ O isotopes are also present. For the species analyzed in this work, these masses do not pose any interferences. It is notable that these could interfere with the calibrations of other species though.

Line 181-183: It is also interesting that, in general, you see higher sensitivity at lower E/N (Section 3.2). It is likely that the furanoids are reactive towards the (H2O)H3O+ cluster (similar to some oxygenates, like acetone) and higher E/N results favorable energies that enhance the mechanism noted here. The PA for the water dimer is ~801 kJ/mol (Yuan et al. 2016). The PA for furfural is ~850 kJ/mol (https://pubmed.ncbi.nlm.nih.gov/23147827/). So it's quite possible that furfural (and probably the other oxygenated furanoids) are reacting with the water cluster and resulting in unique product distributions. This should be noted as a possible explanation at lines 213-215.

Thank you for this possible explanation. We have added a sentence to the manuscript noting this reaction. Specifically, "it is likely these furanoids are reactive with the  $H_2O(H_3O)^+$  cluster (proton affinity 801 kJ mol-1, Yuan et al., 2016), which is more favorable at lower E/N conditions."

Table 1: Suggest changing "Fragments" to "Fragments and other products"

We have made this change to Table 1 according to the suggestion.

Line 261-263: Perhaps some of these differences could be attributed to reactions with the water cluster, as noted above.

This is a good point, and we have noted this as a possible explanation.